# Worry about the Future in the Climate Change Emergency: A Mediation Analysis of the Role of Eco-Anxiety and Emotion Regulation

**DOI:** 10.3390/bs14030255

**Published:** 2024-03-21

**Authors:** Luisa Orrù, Federica Taccini, Stefania Mannarini

**Affiliations:** 1Department of Philosophy, Sociology, Education, and Applied Psychology, Section of Applied Psychology, University of Padua, 35131 Padua, Italy; federica.taccini@unipd.it (F.T.); stefania.mannarini@unipd.it (S.M.); 2Center for Intervention and Research on Family Studies—CIRF—Department of Philosophy, Sociology, Education, and Applied Psychology, Section of Applied Psychology, University of Padua, 35131 Padua, Italy

**Keywords:** eco-anxiety, clinical intervention, worry about the future, mental health, emergency, emotion dysregulation, climate change worry, psychological effects of climate change

## Abstract

The climate change emergency is one of the most important challenges of our time, and its impact on mental health has been evident for years. It is very important for clinicians to delve deeper into these manifestations. There are a wide variety of constructs, symptoms, and scales to measure the impact of climate change on mental health. Eco-anxiety is one of the constructs that has specifically emerged, in association with worry, about the future. In mental health studies, it is important to explore the relationship between eco-anxiety and emotion regulation and how much this relationship leads to worry about the future in order to be able to offer clinical intervention recommendations. Indeed, the hypothesis of this study is that it is possible to develop worries about the future when symptoms of eco-anxiety and a poor ability to regulate emotions are present. Particularly, emotion dysregulation could increase eco-anxiety symptoms and generate worries about one’s own future. For this reason, we have chosen to explore the relationships between these three constructs with the use of a mediation analysis. For this research, 351 participants were recruited in Italy. The proposed mediation model highlighted the findings that emotion dysregulation was positively related with eco-anxiety and that eco-anxiety predicts worry about the future. An association between emotion dysregulation and worry about the future was present. Eco-anxiety appeared to be an important mediator between emotional dysregulation and worry about the future. Emotion regulation could play a pivotal role in addressing concerns about the future. These findings could pave the way for exploring new research avenues and potential clinical interventions.

## 1. Introduction

In recent years, numerous studies have emerged concerning the psychological effects [1,2,3] of emergencies affecting our planet (e.g., war, climate change, economic crises, etc.). The need has therefore arisen for psychologists, psychotherapists, and health services to confront new forms of health manifestations [4] and clinical frameworks. Concerning climate change, in addition to post-traumatic stress disorder (PTSD) following a catastrophic event, eco-anxiety [5] has also been identified as existing in the absence of a specific event. Eco-anxiety has been one of the psychological effects of the climate change emergency and is related to difficult emotion regulation and climate change worry [6,7]. Indeed, data from the literature have suggested worse self-reported mental health states, functional impairment, and an association between eco-anxiety and symptoms of depression, anxiety, stress, insomnia, and reluctance to have children, especially in younger generations [8].

Various definitions of eco-anxiety have been analyzed by Pihkala [9], and he claimed that it was necessary to have more discussions about such definitions. Eco-anxiety is understood to be linked to feelings of anxiety, worry about the future, and apprehension regarding a threat characterized by considerable uncertainty. It has also been considered as distress caused, to a large extent, by the ecological crisis, but it could not be defined as a clinically understood psychological disorder [5,8,10]. It is characterized by symptoms of generalized anxiety, and it manifests in “practical anxiety” [9], linked to problem-solving attitudes. Climate stress factors are “likely to increase the risk of developing mental health problems, particularly in more vulnerable individuals such as children and young people, who often face multiple life stressors without having the power to reduce, prevent, or avoid such stressors” [8], p. 871. According to this point of view, many scholars from different disciplines have studied people’s experiences of changing environmental conditions and their psychological effects [8,11,12,13]. Indeed, as Thoma et al. [14] have underlined, the present global crisis framework can engage psychological defenses, and it has provoked profoundly distressing emotional reactions [15]. Specifically, findings have emerged indicating that young people suffer from eco-paralysis, uncertainty about the future, apathy, loss of control, and powerlessness [16,17], and they may hesitate to try to have children because of the ecological crisis; having a child evokes deep emotions about the future [18].

About the definition of the construct, as Pihkala [9] clearly highlighted, it is necessary to delve deeper into what is meant by anxiety when it comes to climate change, since this very construct is decisive with respect to all the terminology used by scholars. Eco-anxiety has been studied in relation to several emotions (guilt, grief, nostalgia, despair, anger, and so on) by different authors. Despite this, Pikkala [9] underlined how there still is a need for further investigation with the contribution of interdisciplinary research.

### 1.1. The Relationship between Eco-Anxiety and Worry

In general, eco-anxiety has not been officially classified as a pathological disorder; nevertheless, the repercussions of this clinical manifestation have underscored intense levels of eco-worry that may precipitate mental health challenges. Echoing the words of Brophy et al. [19], the authors highlighted the uniqueness of eco-anxiety, attributing it to the awareness of global climate change and the distressing emotions associated with it, independent of any directly experienced traumatic event. People with strong levels of eco-anxiety also reported that “uncertainty about the future was hard to bear and they wrestle with feelings of helplessness in the face of global ecological problems” [9], p. 8. In this regard, the relevance of worry about the future emerged, and it appeared to be an important component of eco-anxiety; indeed, worry about the future highlights the distressing aspect of perceiving the future as uncertain [8]. However, worry has been considered less overwhelming than anxiety, and Ojala showed that there were productive forms of worry and fear related to eco-anxiety [20].

### 1.2. How to Deal with Worry about the Future through Psychological Interventions?

Concerning the title question, it could be considered that on the topic of climate and environmental emergencies, the recommendation of experts from various disciplines, including psychologists, is to develop and enhance patients’ emotion regulation abilities. Indeed, it has been emphasized that, in a clinical setting, it is indispensable to improve patients’ ability to regulate their emotions related to the feeling of uncertainty and to the worry about the future in relation to the climate emergency. Emotion regulation has become important in the emergency field of research on climate change, as some authors have suggested [21,22,23]. Ojala [20,24] has argued that there are different shades of ecological concern and anxiety, underlining the importance of “critical emotional awareness” (CEA); she recognized the value of emotions in eco-anxiety analysis, taking into consideration the various aspects of each emotion. In general, according to some authors [25,26], emotion regulation has become increasingly important in psychological health studies, so much so that it has been conceptualized as an essential characteristic of good functioning [27]. Emotion regulation has been defined as a complex construct concerning abilities to understand and regulate emotions and feelings [28,29]. When these abilities are lacking, we refer to it as emotion dysregulation [28]. Emotion dysregulation can have serious implications for health, especially when linked with anxiety and worry.

### 1.3. The Relationship between Emotion Regulation and Worry

The relationship between emotion regulation and worry has been studied by several scholars. In this regard, research by Zlomke et al. [30] has shown that cognitive emotion regulation strategies play an important role in the management of anxiety and worry for both men and women. Additionally, the relationship between worry and emotion regulation was also explored by Mennin et al. [31] in patients with anxiety. The authors pointed out that patients’ difficulty in understanding emotional experience and their poor ability to modulate emotions increase generalized anxiety disorders; they might use worry as a defensive strategy to control emotional experience. The authors also suggested adopting an emotion regulation perspective for treatment to help patients feel healthier when undergoing an anxiety-like emotional experience.

Thus, emotion regulation appears to play a role in the generation of worry about the future in the presence of eco-anxiety, and the literature suggests exploring the role of emotion regulation. In line with these studies, in this research, it is very important to specifically explore the relationship between eco-anxiety and emotion regulation since it may contribute to the development of worry about the future. These results can inform the development of clinical treatment recommendations. The hypothesis of this study is that with symptoms of eco-anxiety and a poor ability to regulate emotions, it is possible to develop worries about the future regarding aspects related to one’s life plans. For this reason, we have chosen to explore the relationships between three constructs with a mediation analysis.

## 2. Materials and Methods

### 2.1. Procedure

This research consisted of a cross-sectional study [32]. The data were collected in Italy. This research involved the recruitment of 351 people from the general population. However, 8 participants decided not to continue, resulting in a final sample of 343 participants. The recruitment of participants involved reaching out to individuals from the general population by posting and sharing advertisements on various social media platforms such as Facebook and Instagram. The data were collected from February 2023 to July 2023. This process utilized the snowball sampling method [33].

Inclusion criteria for participants were the following: (a) age above 18; and (b) native Italian speakers, declared by participants at the beginning of the questionaire. Exclusion criteria consisted of (c) difficulties in completing the assortment of self-reports due to illiteracy; (d) cognitive and/or visual difficulties; and (e) completing the assortment of questionnaires in less than eight minutes.

This research involved participants who willingly volunteered and expressed their informed consent via approved consent forms with a specific part of questionnaire added in the Qualtrics platform. This platform can save data and then delete them once the collection is complete. The platform takes compilation time into account. In this way, questionnaires compiled in less than 8 min were eliminated. Respondents were informed of their right to decline to participate and to withdraw from the research once participation began as well as whom to contact for questions about the research or in case of any discomfort in answering the questions. Confidentiality was also granted by assigning an anonymous code for each participant, as well as by storing and disposing data securely. Approval for the research project was granted by the Ethics Committee of Psychological Research at the University of Padua (protocol number 5231, 20 February 2023).

### 2.2. Participants

The ultimate group of participants included 343 participants (Mage = 29.91, SD = 13.437). Of these, 252 (73.5%) were female, and 91 (26.5%) were male. Concerning the marital status, 276 (80.5%) were single, and 67 (19.5%) were married, and regarding the level of education, 1 had completed elementary school (0.3%), 5 had completed lower secondary school (1.5%), 154 (44.9%) had completed upper secondary school, 152 (44.3%) were undergraduates, and 31 (9.0%) were post-graduates.

### 2.3. Measures

Demographic data were gathered, encompassing age, gender, marital status, and the level of education. Three different self reports were addimistrated: Difficulties in Emotion Regulation Scale—Short Form, the Hogg Eco-Anxiety Scale, and a questionnaire on worry about the future.

#### 2.3.1. Difficulties in Emotion Regulation Scale—Short Form (DERS-SF)

The Short Form of the Difficulties in Emotion Regulation Scale (DERS-SF) [28,34] was used. This questionnaire comprises 18 items and employs a 5-point Likert-type scale, ranging from “almost never” to “almost always”. The DERS-SF encompasses both a total scale (α = 0.910) and six subscales. Higher scores on this scale indicate a higher frequency of challenges in emotion regulation. The Italian version of the DERS-SF for adults was utilized in this research [35] with the total scale.

#### 2.3.2. Hogg Eco-Anxiety Scale (HEAS)

The Hogg Eco-Anxiety Scale [HEAS, 5] is 13-item questionnaire designed to assess climate change anxiety. This includes evaluating its cognitive, behavioral, and emotional dimensions, along with a sense of personal accountability for climate change. The questionnaire asks participants how often they experienced eco-anxiety symptoms in the previous two weeks (e.g., “Unable to stop thinking about future climate change and other global environmental problem”; “Feeling anxious about your personal responsibility to help address environmental problems”). The HEAS offers a succinct and thorough overview of the diverse expressions of eco-anxiety, which encompasses self-evaluation of one’s role in the repercussions of specific climate-related events. Respondents provide their feedback using a 4-point Likert scale, ranging from 0 (“not at all”) to 3 (“nearly every day”), reflecting on the past two weeks. As conducted in previous studies, the total scale of this measure was used. The internal reliability of the total scale was good, with α = 0.897. The HEAS has been validated for use in an adult Italian population [36].

#### 2.3.3. Worry about the Future

The questionnaire designed to assess worry about the future is a modified version of Hickman et al.’s [8] questionnaire; 6 out of the 8 items were developed using a 4-point Likert scale, ranging from 0 (“not at all”) to 3 (“very much”). The items that were retained were “I am hesitant to have children”; “Humanity is doomed”; “The future is frightening”; “I won’t have access to the same opportunities that my parents had”; “My family’s security will be threatened (e.g., economic, social, or physical security)”; and “People have failed to take care of the planet”. These items focuses on the dimension of worry that arises from climate change; they are representative of the worry construct conceptualized in this project. The questionnaire presents a total scale that shows a good internal reliability: α = 0.85.

### 2.4. Data Analysis

Rstudio [37,38] was used to conduct all the statistical analyses. The following packages were used: for the mediation analysis, the lavaan package was used [39,40]; for Cronbach’s alpha, the GPA rotation was used [41]; for data manipulation and descriptive statistics, the psych [42], tidyverse [43], magrittr [44], and dplyr [45] packages were used.

Before confirming the mediation model, initial analyses were conducted. Descriptive statistics are present in Table 1 [46,47]. To identify potential cases of multicollinearity, correlation analyses were executed using Spearman’s correlation coefficient, particularly suitable for non-normally distributed variables. According to the literature [46,48], correlation coefficients exceeding |0.80| were considered indicative of multicollinearity.

A mediational analysis was performed utilizing observed variables. The direct relationship of a predictor (Emotion Dysregulation) on an outcome variable (Worry about the Future) was investigated. Additionally, this study explored the indirect relationship of Emotion Dysregulation on Worry about the Future through Eco-Anxiety (Figure 1). The maximum likelihood (MLM) estimator was used, and the goodness of fit of the model to the data was assessed using the Satorra–Bentler scaled chi-square test, χ^2^ [47].

#### 2.4.1. Sample Size Determination

The sample size for this study was pre-determined based on the statistical analysis planned for this study, which are outlined in the Data Analysis section. This study employed the “n:q criterion”, as defined by Kline [49], where “n” denotes the number of participants and “q” represents the number of free model paths. To satisfy the minimum sample size criterion, it is recommended to maintain a ratio of no less than 30 participants for each free path (i.e., a 30:1 ratio) [49]. As a result, a total of 90 participants were needed for this research.

#### 2.4.2. Preliminary Results

The correlation analyses, as indicated in Table 1, demonstrated significant relationships among the variables under examination in the mediation analysis. However, none of the correlations exceeded the suggested threshold of |0.80|, indicating that it was appropriate to proceed with the statistical analyses.

**Table 1 behavsci-14-00255-t001:** Descriptive statistics and Spearman’s correlation among variables.

		Descriptive Statistics						
		M	SD	SK	K	1	2	3
1	Emotion Dysregulation	41.04	38.380	0.611	−0.242	-		
2	Eco-Anxiety	8.30	6.945	1.118	1.654	0.365 **	-	
3	Worry about the Future	17.07	4.148	−0.604	−0.064	0.304 **	0.552 **	-

Note: ** *p* < 0.01. M—mean; SD—standard deviation; SK—skewness; K—kurtosis.

The mediation model displayed positive indicators of goodness of fit. The model exhibited good goodness-of-fit indices: Satorra, A., and Bentler scale (χ^2^SB = 0.002) [47] *p* = 0.967; Comparative Fit Index (CFI) = 1 [50,51]; Root-Mean-Square Error of Approximation (RMSEA) = 0.000 [52,53]; Standardized Root Mean Square Residual (SRMR) = 0.001 [54].

In relation to the mediation model (refer to Table 2 and Figure 2), the “Emotion Dysregulation” predictor (X) was positively related to “Eco-Anxiety” (M), path a1: β = 0.195 (SE = 0.031), *p* < 0.001, 95% CI [0.134;0.255]; β* = 0.359. The degree of explained variance was 12.9% (R^2^ = 0.129).

In addition, “Eco-Anxiety” (M) statistically significantly predicted “Worry about the future” (Y), path b1: β = 0.282 (SE = 0.030), *p* < 0.001, 95% CI [0.223;0.341]; β* = 0.472. Furthermore, after controlling for “Eco-Anxiety” (M), there was a statistically significant relationship between “Emotion Dysregulation” and “Worry about the future” (path c1: β = 0.043 (SE = 0.016), *p* < 0.007, 95% CI [0.012;0.075]; β* = 0.134). The degree of explained variance was 28.7% (R^2^ = 0.287).

The total indirect effect was β = 0.055 (SE = 0.009), *p* < 0.001, 95% CI [0.038;0.072]. Moreover, the total effect was β = 0.098 (SE = 0.016), *p* < 0.001, 95% CI [0.066;0.130].

**Figure 1 behavsci-14-00255-f001:**
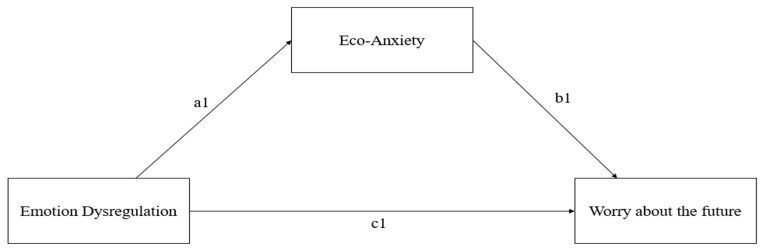
Model conceptualization.

**Figure 2 behavsci-14-00255-f002:**
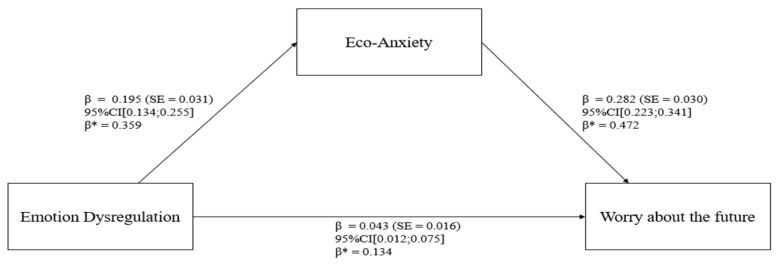
Statistical model.

**Table 2 behavsci-14-00255-t002:** Mediation model coefficients.

Path		β*	β (SE)	95% CI [L-U]	z-Value	R^2^
▪ Outcome: Eco-Anxiety (M) Emotion Dysregulation (X) → Eco-Anxiety (M)	(a1)	0.359	0.195 (0.031)	[0.134;0.255]	6.277 ***	12.9%
▪ Outcome: WF (Y)	(b1)	0.472	0.282 (0.030)	[0.223;0.341]	9.348 ***	28.7%
Eco-Anxiety (M) → WF (Y)						
Emotion Dysregulation (X) → WF(Y)	(c1)	0.134	0.043 (0.016)	[0.012;0.075]	2.690 **	
▪ Effect of X on Y via M	(a1*b1)	0.170	0.055 (0.009)	[0.038;0.072]	6.257 ***	
▪ Total Effect of the Model		0.304	0.098 (0.016)	[0.066;0.130]	6.049 ***	

Note: *** *p* < 0.001; ** *p* < 0.01; WF—Worry about the future.

### 2.5. Discussion

Several studies showed a strong relationship between climate change worry, eco-anxiety, and emotion regulation [9,10,14,20,55,56,57,58]. Therefore, the relationship between eco-anxiety and emotion regulation [35] was investigated. Emotion regulation corresponds to the individuals’ ability to identify and regulate the emotions they have. As many studies available in the literature have focused on the construct of coping [21] (which, despite having similarities with the process of emotional regulation, is not comparable to it), this contribution aimed to focus the attention on the emotional part that influences the management of eco-anxiety and worry about the future, as well as on the literature [30,32] that has highlighted the relationship between anxiety and worry.

In this study, worry about the future was measured with a scale composed of 6 items taken from the study by Hickman et al. [8] and modified with respect to the number of items. The worries explored in the research included the possibilities of having children, the idea that humanity is doomed, how the future is perceived as frightening, the possibility of accessing the same opportunities that the parents had, the family’s security (e.g., economic, social, or physical security), the perception of not adequately caring for the planet, and the relationship between eco-anxiety and emotion regulation and how much this relationship contributes to the development of worries about the future.

This study demonstrated significant relationships among the variables under examination in the mediation analysis. Indeed, the data emerging from the analyses confirmed the hypothesis of this study; an association between eco-anxiety and emotion regulation was found, and it is also clear that this association could lead to a worry about the future. The model suggested a direct effect of emotion dysregulation, as a predictor, on worry about the future as an outcome variable. Additionally, this study explored the indirect impact of emotion dysregulation on worry about the future through eco-anxiety. Symptoms of eco-anxiety might be more pronounced in individuals who lack emotional regulation skills. This could exacerbate worries, especially concerning one’s future plans.

In particular, when examining the relationships among the three constructs identified in the mediation model, eco-anxiety may be viewed as a dimension heightened by emotional dysregulation. Furthermore, emotional dysregulation could exacerbate an individual’s worries, particularly concerning their future plans. These concerns might also be influenced by the individual’s eco-anxiety. Thus, it is conceivable that individuals with strong emotion regulation skills can effectively manage their eco-anxiety and reduce the likelihood of experiencing future-related worries. This perspective holds implications for clinical practice [53,56]; the relevance of paying attention to the constructs of eco-anxiety and worry about the future in the clinical field has been suggested. Furthermore, greater levels of emotion dysregulation could be associated with greater levels of eco-anxiety and worry about the future. In those with high levels of emotion dysregulation and high levels of worry, eco-anxiety may reach levels that deteriorate daily functioning, predicting worse mental health outcomes [59,60,61]. Considering the results, eco-anxiety could be configured as the cornerstone in the treatment of worry about the future. An increase in both eco-anxiety and emotional dysregulation may entail a heightened acknowledgment of the urgency surrounding environmental issues, leading to worries about one’s life goals. To effectively manage eco-anxiety in the presence of emotional dysregulation and to preempt worries about the future, an approach that integrates multiple facets could prove beneficial. This approach involves comprehensively understanding the specific triggers of eco-anxiety and the nuances of emotional regulation through dedicated assessment; employing emotional regulation techniques and teaching patients strategies for regulating and managing their emotions; and encouraging tangible actions that channel worry into environmental activism, thereby fostering a sense of control and contributing to community well-being.

#### Limitations

This study contended with some limitations; first of all, it was conducted with a non-clinical sample. Future research could analyze the association between worry about the future and emotion regulation in groups of patients with eco-anxiety. Moreover, the most important literature research about eco-anxiety and worry regards young people and young adults; in line with these, future studies could test the mediation model specifically with these sample of participants. Another limitation is related to the variables “gender” and “geographic area”. Regarding gender, more women than men participated in the research. Concerning geographic area, the participants came from Italy, but it could be interesting to explore how the model changes with respect to origins from specific geographical areas (with the presence or absence of climatic events, urban or rural areas, etc.).

In the end, this study used an online convenience snowball sampling method [62]. It is therefore not representative of the general population, and as such, any generalization should be made with caution. These limitations could indicate interesting directions for future research.

### 2.6. Conclusions and Future Perspectives

This study explored the relationship between eco-anxiety and emotion regulation and how much this relationship leads to worry about the future. From the tested mediation model, the literature findings were confirmed: eco-anxiety could be linked to emotion dysregulation and to worries about the future. This meant that as emotion dysregulation increased, eco-anxiety also increased.

Thus, considering the results that emerged, psychological treatments should not only explore worry about the future but also (and rather) focus on patients’ strategies for managing emotions related to the anxiety concerning climate change.

About future perspectives, given what emerged from the research, the questions for further studies could be as follows: To what extent could a rise in eco-anxiety and emotional dysregulation translate into an excessive level of worry about one’s future, so much so as to prevent its development? To what extent can this association generate levels of helplessness in managing daily activities rather than the adoption of sustainable behaviors as a means of coping? Considering the results from this study, it could be useful to conduct an in-depth exploration of the data through the analysis of subscales that measure the constructs of eco-anxiety and emotional regulation. It could be essential for the purposes of clinical classification to know the levels of awareness and management of emotions with respect to the objectives and strategies implemented in relation to the specific HEAS scales.

Furthermore, given the results obtained, it would be beneficial to test the mediation model presented in this study with variables related to gender and participants’ geographical location, either urban or rural, as well as areas affected by catastrophic climate events. In addition, conducting longitudinal research in this direction could enrich the research on eco-anxiety. Finally, it would be intriguing to replicate the study within climate activist groups to examine whether emotional dysregulation differentiates activists from non-activists. This exploration could shed light on whether emotion dysregulation contributes to heightened levels of eco-anxiety compared to general concern for the future or if, as the literature suggests, the latter can evolve into proactive behavior beneficial for the individual’s well-being, leading to a reduction in eco-anxiety levels.

## Data Availability

The data presented in this study are available on request from the corresponding author due to privacy.

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
