# Peer review of "Worry about the Future in the Climate Change Emergency: A Mediation Analysis of the Role of Eco-Anxiety and Emotion Regulation"

_behavsci, 2024, doi:10.3390/bs14030255_

Round 1

Reviewer 1 Report

Comments and Suggestions for Authors

1. Please recosider very long paragraphs in the manuscript. Long paragraphs are undesirable.

2. Please avoid using inappropriate self-citations, references 40, 50, 51 and 57. This issue strongly decreases the quality of your paper. Please delete these references.

3. Please indicate date of etchical approval.

The paper is in general good, however this issue of self-citations is very dissapointing.

Some minor comments which can be resolved during editing process:

1.Line 40. Pikhala is a man, so, not "she", but "he".

2. Please add more keywords to improve indexing of the paper.

3. Line 138: not Febrary, but February.

4. Line 172: "The Hogg Eco-Anxiety Scale [HEAS, 5] is a survey ". Please change "a survey" into "a questionnnaire" as the HEAS is a questionnaire.

5. Please reconsider one-sentence paragrpahs. One-sentence paragraphs are undesirable.

6. Table 1: Please decipher abbreviations of SK and K in the notes.

7. Figure 2: Typos in the title.

Author Response

REV_1_1. Please reconsider very long paragraphs in the manuscript. Long paragraphs are undesirable.

Response: Thank you for your feedback. Some paragraphs were added in Introduction.

REV_1_2. Please avoid using inappropriate self-citations, references 40, 50, 51 and 57. This issue strongly decreases the quality of your paper. Please delete these references.

Response: Thank you for your feedback. The self-citations (40, 51, 57) were deleted as you suggested. The 50 (now 49) was used for statistical measure and for this reason it needs to be maintained.

REV_1_3. Please indicate date of ethical approval.

Response: Thank you for your feedback. The date was indicated in “Institutional Review Board Statement” section

REV_1_4: The paper is in general good, however this issue of self-citations is very dissapointing.

REV_1_5: Line 40. Pikhala is a man, so, not "she", but "he".

Response: Thank you for your feedback. We have changed the pronoun.

REV_1_6: Please add more keywords to improve indexing of the paper.

Response: Thank you for your feedback. We have added and changed key-words: clinical treatment; worry about the future; mental health; emergency; emotion dysregulation; emotions in climate change time; psychological effects of climate change.

REV_1_7: 3. Line 138: not Febrary, but February.

Response: Thank you for your feedback, the changed was been made.

REV_1_8: Line 172: "The Hogg Eco-Anxiety Scale [HEAS, 5] is a survey ". Please change "a survey" into "a questionnnaire" as the HEAS is a questionnaire.

Response: Thank you for your feedback. The word questionnaire has been used instead of survey.

REV_1_9: Please reconsider one-sentence paragrpahs. One-sentence paragraphs are undesirable.

Response: Thank you for your feedback. The one-sentence paragraphs have been modified.

Response: Thank you for your feedback,  paragraphs have been modified using your suggestion.

REV_1_10: Table 1: Please decipher abbreviations of SK and K in the notes.

Response: Thank you for your feedback. The abbreviations have been explained in the notes.

REV_1_11: Figure 2: Typos in the title.

Response: Thank you for your feedback, the typo was modified.

Reviewer 2 Report

Comments and Suggestions for Authors

The manuscript entitled worry about the future in climate change emergency: the role 2 of eco-anxiety and emotion regulation. A mediation analysis is an interesting manuscript linking emotions with climate change. The manuscript require minor revision before making the final decision. Here are some comments/suggestions to be considered in the revised version.

Abstract, row 10, the statement regarding climate change threat to the humanity is still there so the authors should use the word “is” instead of was. The information on location of the study (at least country) be provided so the readers can understand about the development level of that specific country because specific location is much important as development level various across continents and countries.

Keywords should include those not given in the title. Presently many words given in keywords are the same provided in the title of the manuscript.

Introduction section starts with the statement “Human health was strongly linked to the natural and anthropic changes affecting 28 our planet (war, climate change, economic crises, etc.), and in recent years numerous studies have emerged concerning their psychological effects” (please see rows 28-9). It implies that humanity was facing different challenges as a result of natural and anthropogenic changes and somehow the world has been able to control those challenges with the passage of the time. And it is not true even in the present time. We see many disasters in the world happening during the whole year. So the authors should be careful in writing these statements in the manuscript.

Rows 49-51, following study be cited to strengthen the statement

Impact of socio-economic and water access conditions on life satisfaction of rural farmers in Faisalabad district of Pakistan. Water Policy, 2020 https://doi.org/10.2166/wp.2020.004

For rows 73-74, the citation to support the statement is lacking.

For row 96-97, climate change has substantial impact on the emotions of humans resulting into different threats to individuals. The manuscript has provided the evidence relating to the impact of climate change on the emotion. 

Material and methods section should include a brief introduction of the study site/location. Map of the study area will be useful in understanding the location. Next section should contain information on population recruitment. Authors need to explain what was the basis for deciding the given sample size? Why snowball sampling technique was chosen? Gender composition is highly skewed in the manuscript. Whether the subjects recruited were taken from rural or urban areas? Writing the statement for example “252 (73.5%) were female and 91 (26.5%) were male” is not good ethically because male and female can be gender of any living organisms i.e. animal, bird, etc. the better way is to write men and women etc.

The first statement of data analysis “Rstudio [38,39,40] was used to conduct all the other statistical analyses” should be rephrased. What are other statistical analysis? I could not find this information in this sub section.

The statement “The sample size for this study was established in accordance with the anticipated 212 statistical analysis outlined in the Data Analysis section” given at 2012-3 is not clear rather confusing.

Further determination of sample size should be taken in the early part of this section.

Figure 2. tatistical model is it a correct or there is something missing or typo error. Statistical or tastical?

Conclusion section usually does not include citation of the literature. Presently it is repetitive of above sections rather it should be concise and brief. It should address findings and implications.

References

Some references are incomplete. These should be carefully written and matched with those given in the text. Journal guidelines be followed in citing and writing references/studies.

For example see reference number 4 in the list, journal issue, volume and page numbers are not provided.

Comments on the Quality of English Language

Minor improvement relating to grammatical and typo errors.

Author Response

REV_2_1: Abstract, row 10, the statement regarding climate change threat to the humanity is still there so the authors should use the word “is” instead of was. The information on location of the study (at least country) be provided so the readers can understand about the development level of that specific country because specific location is much important as development level various across continents and countries.

Response: Thank you for your feedback. We have added the country.

REV_2_2: Keywords should include those not given in the title. Presently many words given in keywords are the same provided in the title of the manuscript.

Response: Thank you for your feedback. We have added and changed key-words: eco-anxiety; clinical intervention; worry about the future; mental health; emergency; emotion dysregulation; climate change worry; psychological effects of climate change.

REV_2_3: Introduction section starts with the statement “Human health was strongly linked to the natural and anthropic changes affecting 28 our planet (war, climate change, economic crises, etc.), and in recent years numerous studies have emerged concerning their psychological effects” (please see rows 28-9). It implies that humanity was facing different challenges as a result of natural and anthropogenic changes and somehow the world has been able to control those challenges with the passage of the time. And it is not true even in the present time. We see many disasters in the world happening during the whole year. So the authors should be careful in writing these statements in the manuscript.

Response: Thank you for your suggestion. We have change rows 33-34 in line with the implications that you highlighted.

REV_2_4: Rows 49-51, following study be cited to strengthen the statement

REV_2_5: Impact of socio-economic and water access conditions on life satisfaction of rural farmers in Faisalabad district of Pakistan. Water Policy, 2020 https://doi.org/10.2166/wp.2020.004

REV_2_6: For rows 73-74, the citation to support the statement is lacking.

REV_2_7: For row 96-97, climate change has substantial impact on the emotions of humans resulting into different threats to individuals. The manuscript has provided the evidence relating to the impact of climate change on the emotion. 

Response: Thank you for your feedback. The citation to support the statement has been added.

REV_2_8: Material and methods section should include a brief introduction of the study site/location. Map of the study area will be useful in understanding the location.

Response: Thank you. The data was collected in Italy. This information has been added in the method section.

REV_2_9: Next section should contain information on population recruitment. Authors need to explain what was the basis for deciding the given sample size?

Response: Thank you for your feedback. The information for deciding the sample size has been reported in the “Sample Size Determination” Section.

REV_2_10: Why snowball sampling technique was chosen?

Response: Thanks to its network characteristics and flexibility, we chose snowball sampling as a recruitment method in order to reach a geographically dispersed population, allowing participants not to feel stigmatized and/or desire anonymity, be particularly sensitive and vulnerable to the topic, obtain a certain degree of confidence to become a willing and participating volunteer.

REV_2_11: Gender composition is highly skewed in the manuscript. Whether the subjects recruited were taken from rural or urban areas? Writing the statement for example “252 (73.5%) were female and 91 (26.5%) were male” is not good ethically because male and female can be gender of any living organisms i.e. animal, bird, etc. the better way is to write men and women etc.

Response: Thank you for your comment. We do not have the information on whether participants were living in a rural or urban area. The mediation model could be improved in next study with the information about their areas information that you suggested (see Limitssection). We changed that sentence to:

LINE 153: “252 (73.5%) were women and 91 (26.5%) were men”

REV_2_12: The first statement of data analysis “Rstudio [38,39,40] was used to conduct all the other statistical analyses” should be rephrased. What are other statistical analysis? I could not find this information in this sub section.

Response: Thank you for your comment and sorry for the typo. That sentence has been rephrased as follows:

LINE 196: “Rstudio [38,39,40] was used to conduct all the statistical analyses”

REV_2_13: The statement “The sample size for this study was established in accordance with the anticipated 212 statistical analysis outlined in the Data Analysis section” given at 2012-3 is not clear rather confusing.

REV_2_14: Further determination of sample size should be taken in the early part of this section.

Response: Thank you for your feedback. That sentence has been rephrased as follows:

LINES 212-218: “The sample size for this study was pre-determined based on the statistical analysis planned for this study which are outlined in the Data Analysis section”.

REV_2_15: Figure 2. tatistical model is it a correct or there is something missing or typo error. Statistical or tastical?

Response: Thank you. Sorry for the typo, the word “tatistical” has been changed to “statistical”

REV_2_16: Conclusion section usually does not include citation of the literature. Presently it is repetitive of above sections rather it should be concise and brief. It should address findings and implications.

Response: Thank you for your feedback. References were deleted and conclusion section re-wrote.

REV_2_17: Some references are incomplete. These should be carefully written and matched with those given in the text. Journal guidelines be followed in citing and writing references/studies. For example see reference number 4 in the list, journal issue, volume and page numbers are not provided.

Response: Thank you for your feedback. References were changed.

Reviewer 3 Report

Comments and Suggestions for Authors

I would like to thank the editors and authors for the opportunity to review the article "Worry about the future in climate change emergency: the role of eco-anxiety and emotion regulation. A mediation analysis"

In general terms, I can say that the article presents specific references in interest, the majority of which are less than 5 years old (more than 66%).

The growing apprehension about the future in the face of the climate emergency has resulted in a notable surge in eco-anxiety among individuals. Although eco-anxiety is an inherent reaction to the climate emergency, it is crucial to adeptly tackle and manage these emotions. The selected theme holds considerable interest and paramount significance, particularly in comprehending the intricate relationships between these constructs through a mediation analysis.

As I will comment below, I have the impression that the manuscript needs some revision and clarification of some aspects.

I will now offer my contributions or suggestions for improving the manuscript:

 1. Introduction

The introduction adeptly highlights the relevance of the study topic with ample substantiation. It effectively defines the problem at hand and succinctly incorporates key insights gleaned from previous investigations, laying a solid foundation for the study.

 2. Materials and Methods

The study's design lacks clear definition of the study design. Additionally, clarity is needed on how the sample size was determined and how missing data were handled. A comprehensive explanation is warranted regarding the measures taken to ensure the anonymity, confidentiality, and privacy of the subjects, particularly highlighting the process of obtaining free and informed consent online.

Furthermore, the authors should specify who had access to the collected data, outline the protective measures implemented, and detail the protocol for data destruction. It is crucial for the authors to elucidate the criteria used to assess participants for cognitive difficulties and to explain the methodology employed to confirm that the battery of questionnaires was completed in under eight minutes.

The statement about participant recruitment involving contact through social media platforms like Facebook and Instagram needs clarification. Specify whether responses were collected online and, if so, elucidate how the informed consent process was conducted using consent forms. If the recruitment process did not occur online, it is advisable to provide a detailed account of the setting, locations, and pertinent dates, encompassing recruitment periods, exposure, follow-up, and data collection phases.

There is a need for the authors to provide a comprehensive description of the scales used, addressing whether they are validated for the specific population under study. Clarity is essential concerning how quotations are interpreted, including the presence of any cut-off points. It is crucial to specify at what point the scores are considered indicative of difficulties in Emotional Regulation, Eco-Anxiety, and Concern about the Future.

Additionally, the items within the Hogg Eco-Anxiety Scale and the Worry about the Future questionnaire require further elucidation. The wording or structure of these items should be clarified to enhance understanding among readers.

The authors assert that "The difference between our study and Hickman's is that the first is based on 6 items and not 7." It is recommended that they provide further clarification on the modification made to the scale, outlining the specific change, the rationale behind this alteration, and whether it underwent validation by experts. This information would enhance transparency and comprehension regarding the adaptation of the scale for their study.

 Results

In both the results and the discussion sections, it is recommended that the authors incorporate a detailed description of the findings, highlighting a comprehensive analysis of the results. Specifically, it is suggested to include the outcomes of the six subscales of the Difficulties in Emotion Regulation Scale – Short Form (DERS-SF). This additional information will provide a more nuanced understanding of the emotional regulation aspects explored in the study.

I would like the authors to clarify the reason for keeping columns 4 5 6 7 8 in table 1 "Descriptive statistics and correlation among variables of correlations", in the correlation section.

 Discussion

The discussion section appears to require further elaboration, necessitating the presentation of results within the broader context of existing literature. Additionally, there is a need for the synthesis of new knowledge generated in response to the research question.

It is recommended to provide a concise summary of the key results with explicit reference to the study's objectives. Furthermore, a thorough discussion on the generalizability (external validity) of the study results is suggested, along with an exploration of the direction and magnitude of potential biases.

 References

The current references show a contemporary focus, with approximately 66% stemming from the last 5 years. It is advised to review the references in accordance with the journal's guidelines, with special attention to details such as reference 13.

 Final Decision:

The manuscript necessitates several noteworthy modifications. I trust that my suggestions will prove beneficial in enhancing the overall quality of this article and the proposed study.

 Thank you very much.

Best Regards

Author Response

In general terms, I can say that the article presents specific references in interest, the majority of which are less than 5 years old (more than 66%).

  1. Introduction

REV_3_1: The introduction adeptly highlights the relevance of the study topic with ample substantiation. It effectively defines the problem at hand and succinctly incorporates key insights gleaned from previous investigations, laying a solid foundation for the study.

  1. Materials and Methods

REV_3_2: The study's design lacks clear definition of the study design.

Response: Thank you for your feedback. A clear definition of the study design has been provided as follows:

LINES 128-129: “The study design of the present research consisted of a cross-sectional study”.

REV_3_3: Additionally, clarity is needed on how the sample size was determined and how missing data were handled.

Response: Thank you for your comment. Information on the sample size determination has been provided in “Sample size determination”. No missing values were present.

REV_3_4: A comprehensive explanation is warranted regarding the measures taken to ensure the anonymity, confidentiality, and privacy of the subjects, particularly highlighting the process of obtaining free and informed consent online.

REV_3_5: Furthermore, the authors should specify who had access to the collected data, outline the protective measures implemented, and detail the protocol for data destruction. It is crucial for the authors to elucidate the criteria used to assess participants for cognitive difficulties and to explain the methodology employed to confirm that the battery of questionnaires was completed in under eight minutes.

REV_3_6: The statement about participant recruitment involving contact through social media platforms like Facebook and Instagram needs clarification. Specify whether responses were collected online and, if so, elucidate how the informed consent process was conducted using consent forms. If the recruitment process did not occur online, it is advisable to provide a detailed account of the setting, locations, and pertinent dates, encompassing recruitment periods, exposure, follow-up, and data collection phases.

Response: Thank you for your comments. We have added the rows below.

LINES 140-150: The present research involved participants who willingly volunteered and expressed their informed consent by approved consent forms with a specific part of questionnaire added in Qualtrix platform. This platform can save data, and then destroy once the collection is complete. The platform takes compilation time into account. In this way, questionnaires compiled in less than 8 minutes have been eliminated. Respondents were informed of their right to decline to participate and to withdraw from the research once participation has begun; and whom to contact for questions about the research or in case of any discomfort in answering the questions. Confidentiality was also granted by assigning an anonymous code for each participant, as well as by storing and disposing data of securely. Approval for the research project was granted by the Ethics Committee of Psychological Research at the University of Padua (protocol number 5231, 20.02.2023).

REV_3_7: There is a need for the authors to provide a comprehensive description of the scales used, addressing whether they are validated for the specific population under study. Clarity is essential concerning how quotations are interpreted, including the presence of any cut-off points. It is crucial to specify at what point the scores are considered indicative of difficulties in Emotional Regulation, Eco-Anxiety, and Concern about the Future.

REV_3_8: Additionally, the items within the Hogg Eco-Anxiety Scale and the Worry about the Future questionnaire require further elucidation. The wording or structure of these items should be clarified to enhance understanding among readers.

Response: Thank you for your comment. The validation details of the questionnaires have been included in the measurement section. We developed the questionnaire regarding worry about the future ourselves. Although one reviewer proposed removing this information from the measurement section, we decided to retain details regarding internal reliability. No clinical cut-off values are present for the questionnaires used for the present research.

LINES 164-169: “The Short Form of the Difficulties in Emotion Regulation Scale (DERS-SF) [30,36] was used. This questionnaire comprises 18 items and employs a 5-point Likert-type scale, ranging from "almost never" to "almost always." The DERS-SF encompasses both a total scale (α = 0.910) and six subscales. Higher scores on this scale indicate a higher frequency of challenges in emotion regulation. The Italian version of the DERS-SF for adults was utilized in this research [37,38] with total scale”.

LINES 171-183: “The Hogg Eco-Anxiety Scale [HEAS, 5] is 13-item a questionnaire designed to assess Climate Change Anxiety. This includes evaluating its cognitive, behavioral, and emotional dimensions, along with a sense of personal accountability for Climate Change. The questionnaire asks participants how often they have experienced eco-anxiety symptoms in the previous two weeks (e.g., “Unable to stop thinking about future climate change and other global environmental problem”; “Feeling anxious about your personal responsibility to help address environmental problems”). The HEAS offers a succinct and thorough overview of the diverse expressions of eco-anxiety, which encompasses self-evaluation of one's role in the repercussions of specific climate-related events. Respondents provide their feedback using a 4-point Likert scale, ranging from 0 ("not at all") to 3 ("nearly every day"), reflecting on the past two weeks. As conducted in previous studies [38], the total scale of this measure was used. The internal reliability of the total scale was good α = 0.897. The HEAS has been validated with adult Italian population [39].”.

 LINES 185-194: The questionnaire designed to assess worry about the future is a modified version of of Hickman et al. [9] questionnaire, 6 out of the 8 items developed using 4-point Likert scale, ranging from 0 (“not at all”) to 3 (“very much”). The items that were retained are: “I am hesitant to have children”; “Humanity is doomed”, “The future is frightening”, “I won't have access to the same opportunities that my parents had”, “My family's security will be threatened (eg, economic, social, or physical security)”, “People have failed to take care of the planet”. These items focuses on the dimension of worry that arises from climate change, they are representative the worry construct conceptualized in this project. The internal consistency results support this choice: the questionnaire presents a total scale that shows a good internal reliability: α = 0.85.  

REV_3_9: The authors assert that "The difference between our study and Hickman's is that the first is based on 6 items and not 7." It is recommended that they provide further clarification on the modification made to the scale, outlining the specific change, the rationale behind this alteration, and whether it underwent validation by experts. This information would enhance transparency and comprehension regarding the adaptation of the scale for their study.

Response: Thank you for your feedback. More information was added. LINES 185-194: The questionnaire designed to assess worry about the future is a modified version of  Hickman et al. [9] questionnaire, 6 out of the 8 items developed using 4-point Likert scale, ranging from 0 (“not at all”) to 3 (“very much”). The items that were retained are: “I am hesitant to have children”; “Humanity is doomed”, “The future is frightening”, “I won't have access to the same opportunities that my parents had”, “My family's security will be threatened (eg, economic, social, or physical security)”, “People have failed to take care of the planet”. These items focuses on the dimension of worry that arises from climate change, they are representative the worry construct conceptualized in this project. The internal consistency results support this choice: the questionnaire presents a total scale that shows a good internal reliability: α = 0.85. 

 Results

REV_3_10: In both the results and the discussion sections, it is recommended that the authors incorporate a detailed description of the findings, highlighting a comprehensive analysis of the results. Specifically, it is suggested to include the outcomes of the six subscales of the Difficulties in Emotion Regulation Scale – Short Form (DERS-SF). This additional information will provide a more nuanced understanding of the emotional regulation aspects explored in the study.

Response: Thank you for your comment. Certainly, information about the outcomes of the six subscales would provide a more nuance understanding of the emotional regulation and we will take this suggestion into consideration for our next study. This one is a cross sectional study to propose a preliminary analysis of the three constructs relationship and so we have focused the attention to the total scale of all the scales. The rows 18-20 and 128-129 have been modified to make the description of the methodological framework consistent with the objectives of the research.

REV_3_11: I would like the authors to clarify the reason for keeping columns 4 5 6 7 8 in table 1 "Descriptive statistics and correlation among variables of correlations", in the correlation section.

Response: Thank you for your comment. Columns 4, 5, 6, 7 have been deleted.

 Discussion

REV_3_12: The discussion section appears to require further elaboration, necessitating the presentation of results within the broader context of existing literature. Additionally, there is a need for the synthesis of new knowledge generated in response to the research question.

Response: Thank you for your comment. Discussion section was improved (see lines: 248-270 and 286-297).

REV_3_13: It is recommended to provide a concise summary of the key results with explicit reference to the study's objectives. Furthermore, a thorough discussion on the generalizability (external validity) of the study results is suggested, along with an exploration of the direction and magnitude of potential biases.

Response: Thank you for your feedback. More information was added (248-270, 310-313 and 286-297).

 References

REV_3_14: The current references show a contemporary focus, with approximately 66% stemming from the last 5 years. It is advised to review the references in accordance with the journal's guidelines, with special attention to details such as reference 13.

Response: thank you, the references were modified.

Reviewer 4 Report

Comments and Suggestions for Authors

Dear authors,

Very interesting manuscript, but I have some recommendations to improve it. It would be useful to formulate the scientific problem in the abstract, which would later lead to clearer conclusions.

I would recommend that you do not list source 37, where it says that it is an article that you are preparing yourself. I would recommend paying attention to the conclusions. The conclusions should reflect the results of your research with a comment on what to do with this information. There really can't be a citation in the conclusion, there's a discussion section for that.

Good luck

Author Response

REV_4_1: Very interesting manuscript, but I have some recommendations to improve it. It would be useful to formulate the scientific problem in the abstract, which would later lead to clearer conclusions.

Response: Thank you for your suggestion. The rows below were added in the abstract.

Line 18-21 Indeed, the hypothesis of the present study was that it is possible to develop worries about the future when symptoms of eco-anxiety and a poor ability to regulate emotions are present. Particularly, an emotion dysregulation could increase eco-anxiety symptoms and generates worries about own future.

REV_4_2: I would recommend that you do not list source 37, where it says that it is an article that you are preparing yourself. I would recommend paying attention to the conclusions. The conclusions should reflect the results of your research with a comment on what to do with this information. There really can't be a citation in the conclusion, there's a discussion section for that.

Response: Thank you for your comment. Source 37 was deleted.

Round 2

Reviewer 1 Report

Comments and Suggestions for Authors

The paper should be reconsidered. It has serious problems, which should preclude its publications in its current form.

1. Please indicate clearly type of correlations? Pearson?

2. Long paragraphs are unwanted (e.g., lines 72-102). One-sentence paragraphs are undesirable.

3. Please correct typos, e.g., "partecipants" etc.

4. Please use abbreviations correctly and decipher them when you introduuce them for the first time, e.g., "GAD".

5. There is a serious problem with references. For example, "Rstudio [39,40] was used to conduct all the statistical analyses".
How are these references related to RStusio?

39. Innocenti, M., Perilli, A., Santarelli, G., Carluccio, N., Zjalic, D., Acquadro Maran, D., ... & Cadeddu, C. (2023). How does climate change worry influence the relationship between climate change anxiety and eco-paralysis? a moderation study. Climate, 11(9), 190. https://www.mdpi.com/2225-1154/11/9/190 448

40. Clayton S, Manning C, Krygsman K, Speiser M. Mental health and our changing climate: Impacts, implications, and guidance. Washington, DC: American Psychological Association and ecoAmerica; 2017. Hinton DE, Nickerson A, Bryant RA. Worry, worry attacks, and PTSD among Cambodian refugees: A path analysis investigation. Soc. Sci. Med. 2011;72(11):1817– 25. doi: 10.1016/j.socscimed.2011.03.045.

6. Please indicate clearly what specific R packages for which specific analyses were used (lines 195-197).

7. I strongly encourage the authors not use innappropriate self-citations in their paper. Innappropriate citations can be treated as an unethical scientific behaviour.

"A mediation analysis was performed utilizing observed variables as conducted in previous study [53,54]".

Reference 53 is inappropriate here. It is definitely not related to the topic. Moreover, it is not a methodological paper which describes mediation analyses.  

53. Taccini, F., & Mannarini, S. (2024). How are survivors of intimate partner violence and sexual violence portrayed on social media?. Journal of Media Psychology: Theories, Methods, and Applications. https://doi.org/10.1027/1864-1105/a000402

8. Lines 227-238: All paths presented in the text can be indicated in the table with mediation analysis. This will increase the readability of the paper.

9.  Casual language is inappropriate in this cross-sectional paper: "The model suggested a direct influence of emotion dysregulation, as a predictor, on worry about the future as an outcome variable. Additionally, the study explored the indirect impact of emotion dysregulation on worry about the future through eco-anxiety."

Impact, affect, influence etc. should be changed.

10. Tautology: "This perspective holds implications for clinical practice.The results of our research could offered various implications for the clinical practice:".

11. The discussion section should be structured in a more concise and logical manner as there are a lot of repetitions of the same things.  

12. "This study explored one of the current emergencies: climate change". This study does not study climate change, but climate change anxiety or people's perceptions of climate change. (For example, exploring the Sun activities is climate change research.).

Author Response

The paper should be reconsidered. It has serious problems, which should preclude its publications in its current form.

  1. Please indicate clearly type of correlations? Pearson?

Response: Thank you for your feedback. This information had been already provided in the paper:

LINES 203-205: “To identify potential cases of multicollinearity, correlation analyses were executed using the Spearman correlation coefficient”

However, in order to address you comment we added this information in the table description as well:

LINE 228: “Table 1. Descriptive statistics and Spearman correlation among variables”

  1. Long paragraphs are unwanted (e.g., lines 72-102). One-sentence paragraphs are undesirable.

Response: thank you for your feedback. The paragraph has been divided into two parts and has been added another one. (see LINE 86)

  1. Please correct typos, e.g., "partecipants" etc.

Response: thank you for your feedback. Sorry for the typos. They have been corrected (see LINE 138 and 308).

  1. Please use abbreviations correctly and decipher them when you introduuce them for the first time, e.g., "GAD".

Response: thank you for your feedback. The abbreviation has been corrected. (LINE 114)

  1. There is a serious problem with references. For example, "Rstudio [39,40] was used to conduct all the statistical analyses".
    How are these references related to RStusio?
  2. Innocenti, M., Perilli, A., Santarelli, G., Carluccio, N., Zjalic, D., Acquadro Maran, D., ... & Cadeddu, C. (2023). How does climate change worry influence the relationship between climate change anxiety and eco-paralysis? a moderation study. Climate, 11(9), 190. https://www.mdpi.com/2225-1154/11/9/190 448
  3. Clayton S, Manning C, Krygsman K, Speiser M. Mental health and our changing climate: Impacts, implications, and guidance. Washington, DC: American Psychological Association and ecoAmerica; 2017. Hinton DE, Nickerson A, Bryant RA. Worry, worry attacks, and PTSD among Cambodian refugees: A path analysis investigation. Soc. Sci. Med. 2011;72(11):1817– 25. doi: 10.1016/j.socscimed.2011.03.045.

Response: thank you for your feedback. The references have been modified (Line 197).

  1. Please indicate clearly what specific R packages for which specific analyses were used (lines 195-197).

Response: Thank you for your comment. That sentence has been rephrased as follows:

LINES 197-201: “The following packages were used: for the mediation analysis the lavaan package was used [42, 43]; for the Cronbach’s alpha the GPA rotation was used [48]; for data manipulation and descriptive statistics the psych [44], tidyverse [45], magrittr [46], dplyr [47] packages were used.

  1. I strongly encourage the authors not use innappropriate self-citations in their paper. Innappropriate citations can be treated as an unethical scientific behaviour.

"A mediation analysis was performed utilizing observed variables as conducted in previous study [53,54]".

Reference 53 is inappropriate here. It is definitely not related to the topic. Moreover, it is not a methodological paper which describes mediation analyses.  

 Response: Thank you. Inappropriate citations were deleted.

  1. Lines 227-238: All paths presented in the text can be indicated in the table with mediation analysis. This will increase the readability of the paper.

Response: Thank you. The names of the paths and of the variables are provided both in the table and in the text.

  1. Casual language is inappropriate in this cross-sectional paper: "The model suggested a direct influence of emotion dysregulation, as a predictor, on worry about the future as an outcome variable. Additionally, the study explored the indirect impact of emotion dysregulation on worry about the future through eco-anxiety."

Impact, affect, influence etc. should be changed.

Response: thank you for your feedback.

“Influence” has been changed with effect:

Line 270: The model suggested a direct effect of emotion dysregulation, as a predictor, on worry about the future as an outcome variable.

“Impact” has been used as a variable, considering the following references. For this reason we would propose to use it in the paper.

  1. Thoma, M., Nicolas R., and Shauna L. Rohner. 2021. Clinical Ecopsychology: The Mental Health Impacts and Underlying Pathways of the Climate and Environmental Crisis. Frontiers in Psychiatry 12: 675936. https://doi.org/10.3389/fpsyt.2021.675936.
  2. Doherty, T. J., & Clayton, S. (2011). The psychological impacts of global climate change. American Psychologist, 66(4), 265.

  1. Tautology: "This perspective holds implications for clinical practice.The results of our research could offered various implications for the clinical practice:".
  2. The discussion section should be structured in a more concise and logical manner as there are a lot of repetitions of the same things.  

Response: thank you for your comment. The discussion section has been modified following your suggestion.

LINE 281-283: This perspective holds implications for clinical practice: on the one hand, they suggested the relevance of paying attention to the constructs of eco-anxiety and worry about the future in the clinical field.

LINE 260-267: In the present study, worry about the future was measured with a scale composed of 6 items taken from the study by Hickman et al., [9] and modified with respect to the number of items. The worries explored in the research included the possibilities of having children, the idea that humanity is doomed, how the future is perceived as frightening, the possibility of accessing the same opportunities that the parents had, the family's security (e.g., economic, social, or physical security), and the perception of not adequately caring for the planet, the relationship between eco-anxiety and emotion regulation and how much this relationship contributes to the development of worries about the future.

Reviewer 3 Report

Comments and Suggestions for Authors

I would like to express my gratitude to the editors and authors for granting me the opportunity to review the article titled "Worry about the future in climate change emergency: the role of eco-anxiety and emotion regulation. A mediation analysis."

After a thorough evaluation, I am pleased to acknowledge that the authors have diligently revised the manuscript in line with the suggestions I provided for improvement.

Consequently, I propose to ACCEPT the article for publication. I extend my congratulations to the authors for their commendable efforts and professional conduct throughout this process.

Best regards,

Author Response

Thank for your feedback.

Round 3

Reviewer 1 Report

Comments and Suggestions for Authors

1. The case of inappropriate self-citations was not resolved, despite the fact that I asked the authors trice to resolve this case of unethical behaviour. This may cause reputational damage for the publisher if the paper is published with unethical case of self-citations.

First case of inappropriate self-citations: Lines 208-209: "A mediational analysis was performed utilizing observed variables as conducted in 208 previous study [52]."

Taccini, F., Rossi, A. A., & Mannarini, S. (2024). Understanding the role of self-esteem and emotion dysregulation in victims of intimate partner violence. Family Process. https://doi.org/10.1111/famp.12966

Second case of inappropriate self-citations: 

The Italian version of the DERS-SF for adults was 170 utilized in this research [37,38] with total scale. Reference 38 is not related to the topic, moreover, DERS-SF was not used in reference 38. 

38. Balottin, L., Mannarini, S., Rossi, M., Rossi, G., & Balottin, U. (2017). The parental bonding in families of adolescents with anorexia: Attachment representations between parents and offspring [Article]. Neuropsychiatric Disease and Treatment, 13, 319- 447. https://doi.org/10.2147/NDT.S128418 

As for the HEAS-13: "As conducted in previous studies [38], the total scale of this measure was used." There is the same problem: HEAS-13 was not used in reference 38. 

The self-citation papers are not methodological papers, and they are not related to the topic of the article. I am extremely confused that the authors did not address this case for three times as in all my three reviews I indicated this problem.  

Third case of inappropriate self-citation: Reference 28 should also deleted for the same reason: Mannarini, S., Balottin, L., Palmieri, A., & Carotenuto, F. (2018). Emotion regulation and parental bonding in families of adolescents with internalizing and externalizing symptoms. Frontiers in Psychology, 9(AUG), Article 1493. 424 https://doi.org/10.3389/fpsyg.2018.01493. 

2. I am also confused the the reference list consists of extremely high amount of errors, with in-text citations referring to inadequate references in bibliography. This was not addressed for the second time (see review report 2).

3. The authors did not reply to comment 12 from the previous review.

4. The fit indices for this mediation model (chi-square, CFI, RMSEA etc.) should be provided as this output was missed (lines 212-214). This is a basic output which should be provided in order to present relevant mediation analysis using SEM.

Author Response

  1. The case of inappropriate self-citations was not resolved, despite the fact that I asked the authors trice to resolve this case of unethical behaviour. This may cause reputational damage for the publisher if the paper is published with unethical case of self-citations.

First case of inappropriate self-citations: Lines 208-209: "A mediational analysis was performed utilizing observed variables as conducted in 208 previous study [52]."

Taccini, F., Rossi, A. A., & Mannarini, S. (2024). Understanding the role of self-esteem and emotion dysregulation in victims of intimate partner violence. Family Process. https://doi.org/10.1111/famp.12966

Response: thank you for your feedback, the reference has been deleted.

Second case of inappropriate self-citations: 

The Italian version of the DERS-SF for adults was 170 utilized in this research [37,38] with total scale. Reference 38 is not related to the topic, moreover, DERS-SF was not used in reference 38. 

  1. Balottin, L., Mannarini, S., Rossi, M., Rossi, G., & Balottin, U. (2017). The parental bonding in families of adolescents with anorexia: Attachment representations between parents and offspring [Article]. Neuropsychiatric Disease and Treatment, 13, 319- 447. https://doi.org/10.2147/NDT.S128418 

As for the HEAS-13: "As conducted in previous studies [38], the total scale of this measure was used." There is the same problem: HEAS-13 was not used in reference 38. 

Response: thank you for your feedback, the reference has been deleted.

The self-citation papers are not methodological papers, and they are not related to the topic of the article. I am extremely confused that the authors did not address this case for three times as in all my three reviews I indicated this problem.  

Third case of inappropriate self-citation: Reference 28 should also deleted for the same reason: Mannarini, S., Balottin, L., Palmieri, A., & Carotenuto, F. (2018). Emotion regulation and parental bonding in families of adolescents with internalizing and externalizing symptoms. Frontiers in Psychology, 9(AUG), Article 1493. 424 https://doi.org/10.3389/fpsyg.2018.01493. 

Response: thank you for your feedback, the reference has been deleted.

  1. I am also confused the the reference list consists of extremely high amount of errors, with in-text citations referring to inadequate references in bibliography. This was not addressed for the second time (see review report 2).
  2. The authors did not reply to comment 12 from the previous review.
  3. The fit indices for this mediation model (chi-square, CFI, RMSEA etc.) should be provided as this output was missed (lines 212-214). This is a basic output which should be provided in order to present relevant mediation analysis using SEM.

Response: Thank you for your comment. The fit indices have been added as follows:

LINES 231-233: “The mediation model displayed positive indicators of goodness of fit. The model exhibited good goodness-of-fit indices: χ2SB(1) = 0.002, p = 0.967; CFI = 1; RMSEA = 0.000; SRMR = 0.001”.

Round 4

Reviewer 1 Report

Comments and Suggestions for Authors

1. Lines 228-230: Abbreviations of the statistical terms (chi square SB, CFI, RMSEA, etc.) were used incorrectly, as they were not deciphered when they were introduced for the first time.  

2. Concerns regarding inappropriate self-citations:

The case of inappropriate citations is still presented.

References 3, 4 and 5 ought to be deleted.

3. Turchi, G. P., Bassi, D., Cavarzan, M., Camellini, T., Moro, C., & Orrù, L. (2023). Intervening on Global Emergencies: The Value of 387 Human Interactions for People’s Health. Behavioral Sciences, 13(9), 735.

This paper is not related to the topic, and it has no words "climate" or similar. 

Turchi, G. P., Orrù, L., Iudici, A., & Pinto, E. (2022). A contribution towards health. Journal of Evaluation in Clinical Practice, 28(5), 389 717. https://doi: 10.1111/jep.13732

Similarly, this paper is not related to the topic, and it has no words "climate" or similar. 

5. Campolonghi, S., & Orrù, L. (2023). Psychiatry as a medical discipline: Epistemological and theoretical issues. Journal of Theoretical 391 and Philosophical Psychology. Advance online publication. https://doi.org/10.1037/teo0000256

Even reading the title, it is possible to realize that this paper is not related to the topic.

In order to prevent reputation damage (Please see consequences of citation manipulation for publishers, authors' institution, and authors here: https://publicationethics.org/citation-manipulation-discussion-document), I extremely strongly recommend the authors to eliminate these inappropriate citations as inappropriate citations are considered as unethical behavior.

Author Response

Lines 228-230: Abbreviations of the statistical terms (chi square SB, CFI, RMSEA, etc.) were used incorrectly, as they were not deciphered when they were introduced for the first time.  

Response: thank you for your feedback. The statistical terms have now been described.

Lines 230-233: The mediation model displayed positive indicators of goodness of fit. The model exhibited good goodness-of-fit indices: Satorra, A., & Bentler scale (χ2SB= 0.002) (49), p = 0.967; Comparative Fit Index (CFI = 1) (50,51); Root-Mean-Square Error of Approximation (RMSEA = 0.000) (52,53); Standardized Root Mean Square Residual (SRMR = 0.001) (54).

  1. Concerns regarding inappropriate self-citations:

The case of inappropriate citations is still presented.

References 3, 4 and 5 ought to be deleted.

  1. Turchi, G. P., Bassi, D., Cavarzan, M., Camellini, T., Moro, C., & Orrù, L. (2023). Intervening on Global Emergencies: The Value of 387 Human Interactions for People’s Health. Behavioral Sciences, 13(9), 735.

This paper is not related to the topic, and it has no words "climate" or similar. 

Response: thank you for your feedback.

The citation was used in the following sentence: “In recent years numerous studies have emerged concerning psychological effects [1,2,3] of emergencies affecting our planet (e.g., war, climate change, economic crises, etc.).”

Although the cited paper does not mention the word “climate” or similar, for the authors this citation is useful to describe the current topic because climate change is one of the main global emergencies of our time. We have considered very important to inscribe climate change emergency within the bigger picture of global emergencies of our time in order to describe how clinicians could treat their psychological effects, that appear to be the same or similar even among different types of emergencies (e.g. covid-19). Indeed, in the cited paper we can find the following sentences:

“In fact, regardless of the peculiar typology of events and effects (contents, different from each other), all emergencies originate from a sudden event that modifies, in a more or less influential way, the community arrangement involved (process).” (p.3)

“Emergency management has been a deeply argued topic in multiple contexts such as terrorism [1,2]), natural disasters [3,4] and public health [5–7]. Public health emergency management has raised as a specific field of practice, since the health impact of infectious diseases, environmental catastrophes, and conflicts in recent years have become increasingly relevant from the perspective of strengthening public health systems and protecting communities from naturally occurring and human-caused threats [8]”. (p.1)

“During the early days of the COVID-19 pandemic, people experienced relevant psychological distress, with symptoms of anxiety, depression, and PTSD observed consistently across the globe [21]. Overall, these behavioral and psychological effects [22–26] plus stress [13,27,28], fear of contagion, death [29,30], and social isolation [31] were the main concerns.” (p. 2)

The complexity of this emergency and its specific criticalities can generate an impact on public health, economics, and people’s mental health on the rest of the world [17–19]. Ref. [20]’s study presented some psychosocial effects of the COVID-19 pandemic among different age groups. Young adults had higher scores in preoccupation and change of habits linked to COVID-19, while older adults were least worried and expressed less fear [20]. (p. 2)

Turchi, G. P., Orrù, L., Iudici, A., & Pinto, E. (2022). A contribution towards health. Journal of Evaluation in Clinical Practice, 28(5), 389 717. https://doi: 10.1111/jep.13732

Similarly, this paper is not related to the topic, and it has no words climate or similar. 

Response: thank you for your feedback.

The citation was used in the following part of our paper: “The need has therefore arisen for psychologists, psychotherapists and health services, to be confronted with new forms of health configuration [4,5] and clinical frameworks”.

The paper cited is useful to the current topic because it describes the health configuration construct (as in the paper’s sentence is used) and the implications for clinicians to treat it. As mentioned, the health configuration construct stands as an ensemble of different areas, ranging from daily events to more widespread issues, such as global emergencies. This paper describes the health configuration construct useful for framing the theoretical-epistemological reference with which the interventions that health services can carry out and orient, as the Special Issue required.

  1. Campolonghi, S., & Orrù, L. (2023). Psychiatry as a medical discipline: Epistemological and theoretical issues. Journal of Theoretical 391 and Philosophical Psychology. Advance online publication. https://doi.org/10.1037/teo0000256

Even reading the title, it is possible to realize that this paper is not related to the topic.

Response: thank you for your feedback. The citation was used in the following part of our paper: “The need has therefore arisen for psychologists, psychotherapists and health services, to be confronted with new forms of health configuration [4,5] and clinical frameworks”.

The cited paper describe the health construct, the disease and normal condition, within an epistemological framework: “It concludes with a discussion of the implications of abandoning psychiatry’s biological framework in mental health care, and the possibility for psychiatry to find its own specific, unique, and legitimate space of knowledge and practice.” (p.1)

“here is no alternative service for people and service users receiving mental health interventions that are as widely recognized and systematized as psychiatry; therefore, they would be deprived of support and validation of their suffering” (p.9)

The cited paper is useful for our research in order to state what is possible to define as Health and clinical contribution. However, we considered that the paper “Turchi, G. P., Orrù, L., Iudici, A., & Pinto, E. (2022). A contribution towards health. Journal of Evaluation in Clinical Practice, 28(5), 389 717. https://doi: 10.1111/jep.13732” is sufficient, and so we prefer to delete this one from the references.